# Can an Ecological Index of Deprivation Be Used at the Country Level? The Case of the French Version of the European Deprivation Index (F-EDI)

**DOI:** 10.3390/ijerph19042311

**Published:** 2022-02-17

**Authors:** Ophélie Merville, Ludivine Launay, Olivier Dejardin, Quentin Rollet, Joséphine Bryère, Élodie Guillaume, Guy Launoy

**Affiliations:** U1086 “ANTICIPE” INSERM, University of Caen Normandy, 14000 Caen, France; ludivine.launay@unicaen.fr (L.L.); olivier.dejardin@unicaen.fr (O.D.); quentin.rollet@inserm.fr (Q.R.); josephine.bryere@unicaen.fr (J.B.); elodie.guillaume@unicaen.fr (É.G.); guy.launoy@unicaen.fr (G.L.)

**Keywords:** deprivation, ecological indices, validation study, area classification

## Abstract

Most ecological indices of deprivation are constructed from census data at the national level, which raises questions about the relevance of their use, and their comparability across a country. We aimed to determine whether a national index can account for deprivation regardless of location characteristics. In Metropolitan France, 43,853 residential census block groups (IRIS) were divided into eight area types based on quality of life. We calculated score deprivation for each IRIS using the French version of the European Deprivation Index (F-EDI). We decomposed the score by calculating the contribution of each of its components by area type, and we assessed the impact of removing each component and recalculating the weights on the identification of deprived IRIS. The set of components most contributing to the score changed according to the area type, but the identification of deprived IRIS remained stable regardless of the component removed for recalculating the score. Not all components of the F-EDI are markers of deprivation according to location characteristics, but the multidimensional nature of the index ensures its robustness. Further research is needed to examine the limitations of using these indices depending on the purpose of the study, particularly in relation to the geographical grid used to calculate deprivation scores.

## 1. Introduction

A large variety of ecological indices of deprivation have been developed around the world to study socio-economic and territorial inequalities [1,2,3,4,5,6,7,8,9,10,11,12,13,14]. They have been widely used in recent years to assess and analyse social inequalities in health [9,14,15,16,17,18,19,20,21,22,23,24,25,26,27,28]. To construct them, it is necessary to define “deprivation”, and to identify indicators that measure it. Townsend [29] defined deprived individuals as those who are not able to meet the needs identified by the majority of the people in the society in which they live. Thus, this definition of relative deprivation and the paths to it may vary according to the time period and the geographical area [29].

Ecological indices typically encompass multiple components of material and social disadvantage to account for the multidimensional aspects of deprivation experienced by residents at the level of a geographical area. Frequently used to compensate for the lack of socio-economic data at the individual level, they include a contextual dimension related to the geographical level of measurement. They allow the assessment of the socio-economic situation of a geographical area by aggregating the socio-economic characteristics of its residents. This contextual dimension has been described in the literature [30,31,32], and corresponds to a part of the “place effect”, which aims to explain the role of location in the construction of health inequalities. Macintyre et al. [32] assumed that “place effects” on health are the result of interrelated compositional and contextual effects. At the neighbourhood level, he emphasised the existence of a social dimension, or “social miasma”, by assuming that being surrounded by disadvantaged people would have an impact—a priori negative—on an individual’s heath regardless of his or her social position. The effect size is variable, and also depends on individual material and social resources.

Ecological indices until now have been developed according to a specific methodology based on the availability of data—which also determines the geographical unit to be used—and prior knowledge of the area. These indices provide deprivation scores which allow the identification of concentrations of deprivation by ranking geographical units in relation to each other.

Most existing ecological indices were developed at the national level from census data [1,2,3,4,5,6]. The relevance of the national level to define deprivation is based on the assumption that the determinants of deprivation are homogeneous within a country. However, a country may be composed of a wide variety of areas with their own cultural and socio-economical specificities. Some authors have suggested that the available national indices are most suitable for identifying deprivation in urban areas, due to their component indicators, and have questioned their use in other types of areas, notably rural areas [33,34,35,36]. Several studies, notably in the UK, have focused on the specificities of deprivation experienced in rural areas in terms of material poverty and social exclusion [36,37,38,39,40]. Fecht et al. [34] suggested that indicators of household fuel poverty, travel time to services, or need for social care could be used to better identify rural deprivation. Specific indices have therefore been developed to measure deprivation exclusively in rural areas [11], but also for other areas with specific characteristics [12,13,14]. These specific indices can provide a more detailed understanding of the deprivation experienced in a given area, but their use is confined to specific studies restricted to strictly homogeneous geographical areas.

National ecological indices are thus necessary for countrywide studies and for intra-national comparisons using administrative units such as counties or departments. However, in order to optimise their use, it is essential to ensure that they are sufficiently flexible to identify deprived areas in different locations [41]. The definition of deprivation on which the index is based, through the variables and weights that compose it, must be broad enough to be relevant across the country for which it has been developed. This is an issue that concerns the research community, but also the decision-makers who use these indices to identify priority areas for intervention.

The European Deprivation Index (EDI) is one of the available national ecological indices of deprivation, and has already been widely used in studies of social inequalities in health [24,25,26,27,28], but also outside the health field [42]. Due to its concept and methods of construction, the EDI is likely to be replicated across Europe, and updated over time. However, despite its theoretical strengths for intra-European comparisons, the EDI shares the concerns expressed about other national ecological indices regarding their adaptability to different specific areas within a country.

Our aim was to determine whether an ecological index defined at the national level is relevant to account for deprivation in a country composed of areas with heterogeneous physical and social environments. Using the French version of the EDI (F-EDI), we explored how the index fits the characteristics of various areas, categorised according to quality of life, in Metropolitan France.

## 2. Materials and Methods

### 2.1. Geographical Scale

The geographical units used were the Ilôt Regroupés pour l’Information Statistique (IRIS), which are the smallest geographical census units available in France. Municipalities with more than 10,000 inhabitants, and most municipalities with 5000 to 10,000 inhabitants are subdivided into several IRIS. In other cases, an IRIS corresponds to a municipality. Each residential IRIS corresponding to a municipal division comprises an average of 2000 inhabitants, as defined by the National Institute of Statistics and Economic Studies (INSEE). We excluded non-residential IRIS, for which there are no F-EDI scores, and those in the Overseas Departments and Territories because the area classification used was not available there. In addition, for the construction of the F-EDI, individuals from these territories were not part of the French sample used to define deprivation, which raises issues about the use of the index outside Metropolitan France. Thus, this study focused on the 47,853 IRIS in Metropolitan France.

### 2.2. Deprivation Measure

The method of EDI construction is the same for each country, based on 3 steps, and using individual data of the European Union Statistics on Income and Living Conditions (EU-SILC) survey and census data aggregated to the smallest geographical unit available. In the first step, country-specific fundamental needs are identified from the EU-SILC survey, leading to the construction of a country-specific individual deprivation indicator. In the second step, the variables common to the EU-SILC survey and the national census are identified and recoded to be similar. In the third step, a multivariate logistic regression is performed to select the variables most highly correlated with the EU-SILC individual deprivation indicator. The regression coefficients are used as weights in the final index formula. The EU-SILC survey and national censuses are frequently repeated, allowing the EDI to be updated over time.

For this study, we used the F-EDI, updated in 2015, to measure deprivation in Metropolitan France. This version of the F-EDI has been constructed from the French sample of the EU-SILC survey (https://www.eui.eu/Research/Library/ResearchGuides/Economics/Statistics/DataPortal/EU-SILC, accessed on 8 February 2022) and national census data, at the IRIS level, produced by INSEE for the year 2015 (https://www.insee.fr/fr/statistiques/3625116?sommaire=3558417, accessed on 8 February 2022). The F-EDI score was calculated for each residential IRIS using the following formula:(1)$$\begin{array}{cc} \mathrm{F-EDI}\; & =0.50\;*\;\mathrm{No}\;\mathrm{access}\;\mathrm{to}\;a\;\mathrm{car}\\ & +\;0.84\;*\;\mathrm{Non-owner}\\ & +\;0.44\;*\;\mathrm{Overcrowding}\\ & +\;0.64\;*\;\mathrm{Low}\;\mathrm{level}\;\mathrm{of}\;\mathrm{education}\\ & +\;0.97\;*\;\mathrm{Unskilled}\;\mathrm{worker}\\ & +\;0.73\;*\;\mathrm{Foreign}\;\mathrm{nationality}\\ & +\;1.11\;*\;\mathrm{Single-parent}\;\mathrm{household}\\ & +\;0.25\;*\;\mathrm{Household}\;\mathrm{with}\;\mathrm{two}\;\mathrm{or}\;\mathrm{more}\;\mathrm{persons}\\ & +\;0.97\;*\;\mathrm{Unemployment}\\ & +\;0.39\;*\;\mathrm{Not}\;\mathrm{married} \end{array}$$

Each component corresponds to the weighted rate of the variable in the IRIS according to 2015 census data. All rates were standardised on the French Metropolitan mean of all residential IRIS. For each IRIS, the ten components contributed deprivation points to obtain the F-EDI score. Each residential IRIS was assigned a rank based on the associated F-EDI score. The most deprived IRIS had the highest scores, and were at the top of the ranking.

### 2.3. Area Classification

To investigate the impact of the location on the experience of deprivation, we sought to go beyond the classic rural–urban distinction. We used Reynard’s classification into 8 types of areas based on a geographical division proposed by INSEE (https://www.insee.fr/fr/statistiques/1281328, accessed on 8 February 2022). This INSEE administrative division provides the smallest geographical areas where inhabitants have access to the most common facilities and services. Areas with more than 50,000 inhabitants are divided into several areas to better reflect the diversity of the quality of life in highly urbanised areas. In 2012, Metropolitan France was divided into 2677 areas. Using this geographical scale, Reynard developed an area classification based on 27 indicators, collected between 2011 and 2013, covering 14 dimensions of quality of life [43]. These dimensions cover most aspects of social life, such as access to facilities, cultural/sport/leisure activities or community life, education, gender equity, employment, environment, work–life balance, accommodation, social relationships, income, health, safety, public transport, and civic life. Thus, 8 types of areas were identified by conducting a principal component analysis followed by a hierarchical ascending classification (Figure 1):-Type 1: Highly urbanised and rather favourable areas, but with social difficulties and jobs that are often far away.-Type 2: Rather favourable areas with rapid access to facilities, but with socio-economic difficulties.-Type 3: Dense and rich suburbs, but with significant gender disparities.-Type 4: Rather well-off areas, but far from employment, mainly located in the suburbs.-Type 5: Rather dense areas in an unfavourable situation.-Type 6: Small towns in an intermediate situation.-Type 7: Remote and sparsely urbanised areas outside the influence of major centres.-Type 8: Areas around medium-sized towns, offering jobs and rather favourable living conditions.

### 2.4. Statistical Analysis

We cartographically represented the distribution of the F-EDI score within Metropolitan France. For each type of area, we calculated the mean F-EDI score at IRIS level and the standard deviation to obtain a measure of inter-IRIS heterogeneity. We observed the heteroscedasticity of the data using the Levene test and residuals plots. We therefore used Welch’s one-way ANOVA test, which is an alternative to the standard one-way ANOVA test in the event of heteroscedasticity, for the overall comparison of means.

We then investigated which set of components contributed most to the F-EDI score according to the type of area. For each type of area, we calculated the mean number of deprivation points contributed by each component to the F-EDI score. We used Welch’s one-way ANOVA test for overall comparisons, and the Games–Howell post-hoc test for multiple pairwise comparisons.

Finally, we created 10 alternative versions of the F-EDI by successively removing one component from the initial version of the index. This change was made in the third step of the index construction: for each alternative version, one variable was not included in the logistic regression model, resulting in a recalculation of the weights. We assessed the changes in IRIS ranking due to the recalculation of the deprivation score with these alternative versions of the index, and then calculated Kendall’s tau-b correlation coefficients to test the robustness of the F-EDI. To further investigate the relationships of the F-EDI variables with each other, we produced correlograms—available in the Appendix A section—representing the correlation coefficient matrices between the F-EDI variables. We performed the analyses for the whole of Metropolitan France, and by type of area, to investigate potential contextual effects.

Data were analysed using R software version 3.5.3 (R Core Team, Vienna, Austria).

## 3. Results

The geographical distribution of F-EDI scores in Metropolitan France is shown in Figure 2.

The 43,853 residential IRIS in Metropolitan France were distributed among the eight types of areas as follows (Table 1): 3466 type 1, 7624 type 2, 431 type 3, 2630 type 4, 6912 type 5, 10,107 type 6, 7304 type 7, and 9379 type 8. A total of 63,723,769 inhabitants were unevenly distributed in the Metropolitan territory, and the mean number of inhabitants per IRIS varied according to the degree of urbanisation of the area, with more than 2000 for the first four types, around 1300 for types 5 and 8, and fewer than 1000 for types 6 and 7.

The mean F-EDI score was equal to 0 at the metropolitan level because of the standardisation step (Table 1). The mean F-EDI scores at IRIS level were significantly different for the eight types of areas overall (*p* < 0.001). Mean F-EDI scores were lowest in areas located in affluent suburbs of large cities (type 3 and type 4, with, respectively, m = −2.48 and m = −2.64). We also observed lower mean F-EDI scores than the metropolitan mean in areas corresponding to small towns and their surroundings (type 6 and type 8, with, respectively, m = −1.08 and m = −1.15) and, to a lesser extent, in remote and sparsely urbanised areas (type 7, m = −0.6). The highest mean F-EDI scores were found in densely populated areas, with good facilities but social difficulties (type 1 and type 2, with, respectively, m = 4.05 and m = 1.69), or in an unfavourable situation (type 5, m = 1.04). It was also in these areas that we observed the most inter-IRIS heterogeneity (type 1, type 2, and type 5, with, respectively, sd = 5.73, sd = 4.95 and sd = 4.08).

The mean contributions of the components to the deprivation score could be positive or negative, and the set of most contributing components varied according to the type of area (Table 2). A positive value meant that a component had a mean contribution to the F-EDI score higher than the metropolitan mean (i.e., 0), and conversely for a negative value.

In highly urbanised areas with good facilities but social difficulties (types 1 and 2), we observed similar trends in the distributions of the mean contributions of the components to the F-EDI score, with more extreme values in the Paris region (type 1). In these areas, all components had positive mean contributions, except for “Low level of education”, “Unskilled worker”, and “Household with two or more persons”, whose mean contributions were negative. Dense and rich inner suburbs (type 3) had a similar pattern, except for “Unemployment” and “Not married”, whose mean contributions were negative and close to 0, respectively. In large, relatively well-off suburbs (type 4), “Low level of education” and “Unskilled worker” also had high negative contributions. The other components had mean contributions that were negative or close to 0, except for “Household with two or more persons”, which had a positive mean contribution. Dense areas in unfavourable situations (type 5) had a completely different pattern, with positive mean contributions of “Low level of education”, “Unskilled worker”, “Unemployment”, and “Household with two or more persons”, whereas the mean contributions of the other components were close to 0. In small towns and their surroundings (types 6 and 8), “Low level of education”, “Unskilled worker”, and “Household with two or more persons” had positive mean contributions, whereas those of the other components were negative. The pattern was similar in the remote and sparsely urbanised areas (Type 7), with a higher mean contribution of “Unskilled worker”.

Using overall comparisons, the mean contributions of each component were significantly different for the eight types of areas (Table 2, *p* < 0.001), whereas similarities were noted using pairwise comparisons (Table 3). For example, the mean contribution of “Household with two or more persons” to the deprivation score was equivalent for dense and rich inner suburbs (type 3, m = −0.06), and remote and sparsely urbanised areas (type 7, m = −0.04).

The 10 recalculated versions of the F-EDI obtained by successively removing each component led to new weighting schemes for the remaining census variables (Table 4).

At the level of Metropolitan France, the IRIS ranking remained stable regardless of the F-EDI version used (Figure 3a). The lowest Kendall’s tau-b correlation coefficient was obtained with the recalculated F-EDI version without “Single-parent household” (K = 0.77), and the highest with the recalculated version without “Overcrowding” (K = 0.95). IRIS ranking was also stable with the recalculated versions of the F-EDI for the eight types of areas (Figure 3b,c). All the Kendall’s tau-b correlation coefficients were higher than 0.75, except with the recalculated F-EDI version without “Single-parent household” for type 6 (K = 0.70) and type 7 (K = 0.63).

Overall, the differences in IRIS ranking between the initial F-EDI version and the recalculated versions was distributed bilaterally (Figure 3). Nevertheless, we observed trends with the removal of some components, especially for type 1 and type 3 areas. In these types, the removal of “Low level of education” and “Unskilled worker” led to an overall increase in ranking, i.e., IRIS were considered more deprived with these recalculated F-EDI versions than with the initial version. The opposite was observed, to a lesser extent, for types 1 and 3 with the recalculated F-EDI versions without “No access to a car” or “Foreign nationality”.

We observed positive or zero correlations between the variables composing the F-EDI, except for “Household with two or more persons”, which showed some negative correlations (Figure A1 and Figure A2). The correlation matrices were different across the types of areas, and there were fewer correlations between the variables in the most rural areas (types 6 and 7).

## 4. Discussion

The F-EDI is composed of a wide range of deprivation indicators that contribute variably to the deprivation score depending on the characteristics of the area. Thanks to its construction methodology, the F-EDI appears to be a robust index for identifying concentrations of deprivation throughout Metropolitan France.

The distribution of the contribution of each component reveals a compositional effect modulated by the weighting associated with each variable. Thus, in remote and sparsely urbanised areas (type 7), the components “Low level of education” and “Unskilled worker” were the main sources of deprivation. This result can be explained by the concentration of high-responsibility positions in areas of dynamic activity. In areas well-served by public transport, such as urban area type 1, a significant proportion of inhabitants did not have access to a car, so this component contributed substantially to the deprivation score.

Our findings lead us to question the indicators composing the F-EDI to define deprivation in Metropolitan France. In the case of the “No access to a car” component, the preference for public transport, where possible, may reflect a choice not determined by economic constraints, especially in areas well-served by public transport. In contrast, in remote areas, car ownership may be considered a necessity, and even the most deprived households need it [44]. Moreover, the choice not to use a car, and to favour alternative and environmentally friendly means of transport, can also be part of a sustainable development approach. The hypothesis that not having access to a car is not a sign of deprivation in some areas is reinforced by the results of the analysis of the differences in IRIS ranking obtained by removing this component from the overall F-EDI score. In the most urbanised areas, the IRIS were almost systematically considered less disadvantaged with the recalculated version of the F-EDI score, whereas we did not observe this trend in the other areas (Figure 3). In all types, it was the removal of “Single-parent household” that led to the most changes in IRIS ranking. However, we did not identify any trends, and it may be assumed that this variable is a marker of deprivation in all types of areas.

The way in which variables in a deprivation index are interpreted according to location is a legitimate concern, and challenges the relevance of an index defined at the national level. However, the present findings demonstrate the robustness of an ecological deprivation index such as the F-EDI, despite the observed trends. Indeed, regardless of the type of area, recalculation of the index after removing each component led to relatively small changes in IRIS ranking (Figure 3). The components with the highest weight, i.e., “Single-parent household”, “Unskilled worker”, and “Unemployment” were logically those whose removal had the greatest impact, although the correlation coefficients showed relative stability across the eight types. One reason for this robustness could be the correlation of the variables with each other, so that no single dimension of deprivation was expressed by a single variable. In addition, the stability of the results when alternative weighting schemes were applied also supports the methodological foundations of the EDI [45,46].

The area classification is a crucial contextual issue, and should be defined according to the research objective [36,37]. Studies on the role of place in defining deprivation have frequently contrasted rural and urban settings [4,16,47,48,49]. The rural/urban dichotomy is classically defined on the basis of population density. Some authors also take the continuity of built-up areas into account, but the threshold for distinguishing a rural area from an urban one is arbitrary. The two categories thus obtained are therefore very heterogeneous, combining major and intermediate cities in one category, and the rest of the territory in the other. The present study highlights, in part, the limitations of this dichotomy for investigating deprivation in different areas. We used an area classification based on quality of life to obtain a more meaningful classification for studying the contextual dimension of disadvantage. We thus observed similar contributions of the components to the deprivation score between different areas, independently of their population density (Table 3). Moreover, the consistency of the distribution of our F-EDI scores with Reynard’s classification tends to demonstrate the reliability of the EDI.

We observed the highest inter-IRIS heterogeneity in the densely populated areas (Table 1), which is consistent with the literature [35,36,50]. Some studies have suggested that the socio-economic characteristics of residents within geographical units in urban areas are more homogeneous than those in rural areas. Owing to their high population density, urban geographical units are smaller and have a greater number of residents than areas remote from centres of activity. The administrative boundaries used to calculate the deprivation score might therefore correspond to neighbourhoods, whereas these divisions might be less meaningful in sparsely populated areas. Several authors refer to “pockets of deprivation” in rural areas, and argue that deprived households are scattered throughout even the most apparently affluent places [33,35,36,50,51]. Thus, the F-EDI might identify concentrations of deprivation that correspond to deprived neighbourhoods, although this geographical grid would be less suitable for remote and sparsely populated areas. It could be assumed that the underestimation of deprivation in rural areas reported in the literature is largely related to the geographic scale used rather than to a lack of specificity of the variables that constitute the deprivation index. Household-level data would be useful for measuring intra-IRIS heterogeneity by area type, and would provide further support for using the F-EDI.

This study has limitations, and further research is needed to improve the understanding and the use of ecological indices of deprivation. First, there was a slight time lag between the data used for area classification (2011 to 2013) and for F-EDI construction (2015), which could involve a timeliness issue. Also, we used an area classification based on quality of life, but we could consider replicating the analyses using other classifications. The use of area classifications based on other geographical divisions, and incorporating other indicators, such as the classification proposed by Fayet et al. [52] for Metropolitan France, would complete this study.

Another approach to assess the relevance of using the F-EDI for the whole Metropolitan France would be to create area-specific F-EDI to compare with the national version. The set of area specific deprivation indicators would, however, be limited by the variables available in the national census and EU-SILC survey, and these specific indices would only be developed for diagnostic purposes.

In order to generalise our results to all national ecological indices of deprivation, these analyses should be replicated with other existing indices. For the EDI, this study should be extended to other national versions of the index. This could provide new arguments for the use of EDI, which already has national versions for France, Italy, Portugal, Slovenia, Spain, and the UK [53].

As a perspective to this study, we could consider evaluating the impact of the removal of each component on the observed differential on established outcomes by type of area. For example, we could assess the differences in mortality or cancer incidence using the overall F-EDI, and compare them to those observed with the 10 alternative versions.

Like most ecological indices of deprivation, the EDI does not include variables relating to the organisation and functioning of a territory, e.g., in terms of infrastructure. These contextual factors, which are defined at different scales, also contribute to the creation of social inequalities in health, and differ according to the issue of interest [30,31,32]. Depending on the research question, using only an ecological index of deprivation to characterise an individual’s socioeconomic environment may not be sufficient. The use of ecological indices within multilevel analyses should be preferred, as soon as the data allow it, as it provides a more detailed understanding of the different mechanisms leading to health inequalities [19,26,49]. By integrating socio-economic indicators at several levels, from the individual to a more or less large geographical area, the individual and contextual effects can be disentangled.

The major advantage of national ecological indices of deprivation is that they provide a uniform measure of deprivation within a country, making intra-national comparisons possible. The EDI extends this advantage to the European level through a common methodological basis, for the construction of national deprivation indices, based on Townsend’s theorisation of relative deprivation. However, depending on the focus of the study, the use of a national index may not be the most relevant. For example, to accurately investigate the mechanisms leading to health inequalities in a specific geographical area, it may seem more relevant to use a specific index to better reflect the reality of the population.

National ecological indices of deprivation are widely used by the research community and by public decision-makers. It is therefore essential that validated measurement tools are available, and that their use does not lead to the invisibilisation of deprivation suffered in some areas, which would be a major source of bias for studies, and have implications for resource allocation [51,54]. All existing ecological indices include subjectivity in their construction methodology due to the choice of variables, weights, and the geographical scale used. Unfortunately, there are still too few validation studies, including robustness studies, to assess these indices. By studying the case of EDI, this study has highlighted the strengths while pointing out the limitations of this type of index to better guide their use.

## 5. Conclusions

It cannot be denied that the experience of deprivation differs slightly according to the location. Nevertheless, the F-EDI seems reliable for identifying concentrations of deprivation across the whole of Metropolitan France, despite the definition of deprivation established at the national level. The F-EDI appears to be a robust ecological index which captures deprivation in different areas, thanks to its component variables and their correlation. The analysis described herein could be applied to other national versions of the EDI to determine whether the findings can be extended to other European countries. Similar findings would support the use of the EDI as an effective statistical tool for studying deprivation thanks to its robustness and cross-cultural dimension. This would feed into policies to tackle social inequalities in health at the European level.

## Figures and Tables

**Figure 1 ijerph-19-02311-f001:**
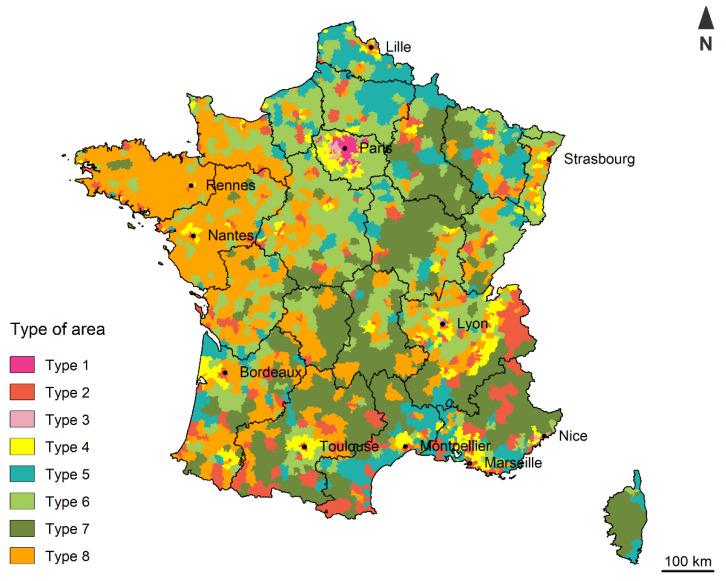
The 8 types of areas in Metropolitan France.

**Figure 2 ijerph-19-02311-f002:**
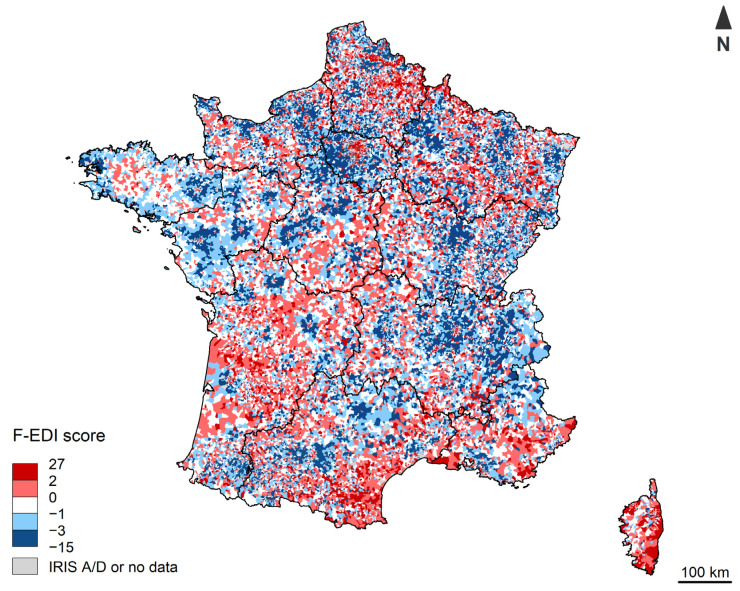
F-EDI score distribution in Metropolitan France.

**Figure 3 ijerph-19-02311-f003:**
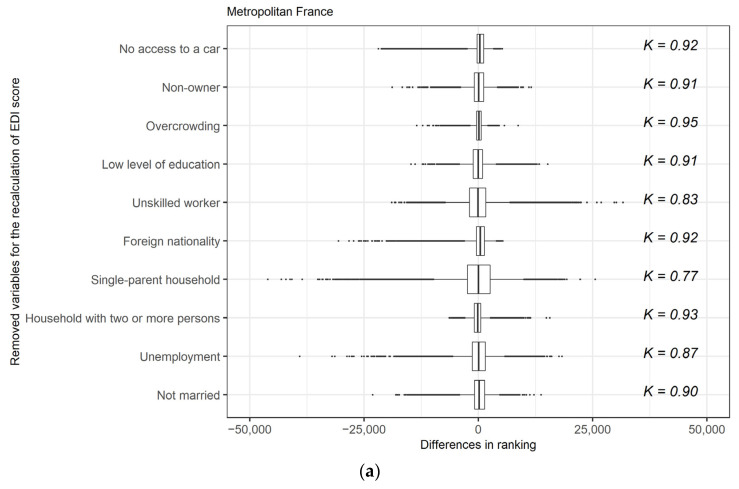
Kendall’s tau-b correlation coefficients and distribution of differences in IRIS ranking by removing components from initial version of F-EDI in Metropolitan France (**a**), and within the eight types of areas (**b**,**c**).

**Table 1 ijerph-19-02311-t001:** Distribution of IRIS and inhabitants, and mean F-EDI scores in the eight types of areas in Metropolitan France.

	IRIS			Inhabitants			
	N	Mean F-EDI Score (sd)	*p*-Value *	n	Mean per IRIS (sd)	Minimum per IRIS	Maximum per IRIS
Metropolitan France	43,853	0.00 (3.92)		63,723,769	1332 (1273)	1	13,202
Type 1	3466	4.05 (5.73)	<0.001	9,121,663	2632 (947)	71	10,337
Type 2	7624	1.69 (4.95)		15,928,400	2089 (1176)	4	9576
Type 3	431	−2.48 (3.13)		1,047,043	2429 (767)	112	8075
Type 4	2630	−2.64 (2.55)		5,363,848	2039 (1422)	31	13,202
Type 5	6912	1.04 (4.08)		9,002,941	1303 (1194)	16	9431
Type 6	10,107	−1.08 (2.42)		8,001,368	792 (984)	3	9922
Type 7	7304	−0.6 (2.56)		2,985,770	409 (625)	1	6572
Type 8	9379	−1.15 (2.47)		12,272,735	1309 (1249)	12	9638

* Welch one-way ANOVA test performed only for the eight types of areas.

**Table 2 ijerph-19-02311-t002:** Mean contributions of components in each of the eight types of areas.

	Metropolitan France	Type 1	Type 2	Type 3	Type 4	Type 5	Type 6	Type 7	Type 8	
F-EDI Score Components	Mean (sd)	Mean (sd)	Mean (sd)	Mean (sd)	Mean (sd)	Mean (sd)	Mean (sd)	Mean (sd)	Mean (sd)	*p*-Value *
No access to a car	0.00 (0.50)	0.97 *^a^* (0.81)	0.29 (0.57)	0.22 (0.50)	−0.23 (0.22)	−0.02 (0.37)	−0.21 (0.19)	−0.14 (0.22)	−0.19 (0.22)	<0.001
Non-owner	0.00 (0.84)	1.13 (0.94)	0.69 (0.98)	0.42 (0.78)	−0.17 (0.59)	0.03 (0.83)	−0.40 (0.46)	−0.36 (0.46)	−0.25 (0.54)	<0.001
Overcrowding	0.00 (0.44)	1.00 (0.72)	0.09 (0.44)	0.30 (0.34)	−0.07 (0.25)	−0.03 (0.33)	−0.12 (0.21)	−0.15 (0.23)	−0.16 (0.16)	<0.001
Low level of education	0.00 (0.64)	−0.68 *^b^* (0.87)	−0.27 (0.71)	−1.53 (0.52)	−0.62 (0.47)	0.28 (0.47)	0.21 (0.44)	0.23 (0.49)	0.11 (0.42)	<0.001
Unskilled worker	0.00 (0.97)	−1.13 (1.18)	−0.20 (0.88)	−1.83 (0.69)	−0.98 (0.70)	0.31 (0.71)	0.13 (0.82)	0.57 (0.86)	0.14 (0.72)	<0.001
Foreign nationality	0.00 (0.73)	1.34 (0.99)	0.24 (0.79)	0.41 (0.47)	−0.11 (0.48)	−0.09 (0.65)	−0.24 (0.46)	−0.13 (0.52)	−0.25 (0.38)	<0.001
Single-parent household	0.00 (1.11)	0.71 (1.08)	0.29 (1.03)	0.11 (0.65)	0.00 (0.73)	0.13 (1.10)	−0.23 (1.07)	−0.23 (1.42)	−0.18 (0.89)	<0.001
Household with two or more persons	0.00 (0.25)	−0.14 (0.28)	−0.20 (0.32)	−0.06 (0.24)	0.12 (0.19)	0.05 (0.20)	0.11 (0.17)	−0.04 (0.20)	0.07 (0.19)	<0.001
Unemployment	0.00 (0.97)	0.53 (1.02)	0.45 (1.23)	−0.49 (0.48)	−0.44 (0.51)	0.43 (1.09)	−0.20 (0.68)	−0.30 (0.92)	−0.29 (0.66)	<0.001
Not married	0.00 (0.39)	0.32 (0.35)	0.30 (0.50)	−0.03 (0.34)	−0.16 (0.30)	−0.04 (0.34)	−0.13 (0.28)	−0.03 (0.35)	−0.12 (0.30)	<0.001

* Welch one-way ANOVA test performed only for the eight types of areas. *^a^* For the IRIS of type 1 area, the mean contribution of “No access to a car” to the F-EDI score was 0.97 points. Taking as reference the mean contribution at the level of Metropolitan France (i.e., 0), this component provided, on average, deprivation points to the IRIS of type 1 area. *^b^* For the IRIS of type 1 area, the mean contribution of “Low level of education” to the F-EDI score was −0.68 points. Taking as reference the mean contribution at the level of Metropolitan France (i.e., 0), this component deducted, on average, deprivation points from the IRIS of type 1 area.

**Table 3 ijerph-19-02311-t003:** Pairwise comparisons of component contributions of the eight types of areas.

F-EDI Score Components	Pairs of Types of Areas with No Significant *p*-Values (*p* > 0.05) *
No access to a car	Type 2 vs. Type 3
Non-owner	
Overcrowding	
Low level of education	Type 6 vs. Type 7
Unskilled worker	Type 6 vs. Type 8
Foreign nationality	Type 4 vs. Type 5/Type 4 vs. Type 7
Single-parent household	Type 3 vs. Type 5/Type 6 vs. Type 7/Type 7 vs. Type 8
Household with two or more persons	Type 3 vs. Type 7
Unemployment	Type 2 vs. Type 5/Type 3 vs. Type 4/Type 7 vs. Type 8
Not married	Type 1 vs. Type 2/Type 3 vs. Type 5/Type 3 vs. Type 7/Type 5 vs. Type 7/Type 6 vs. Type 8

* Games–Howell post-hoc test.

**Table 4 ijerph-19-02311-t004:** Coefficients of variables for the original version and for the 10 alternative versions of F-EDI.

	F-EDI Versions *										
	Full Version	No Access to a Car	Non-Owner	Overcrowding	Low Level of Education	Unskilled Worker	Foreign Nationality	Single-Parent Household	Household with Two or More Persons	Unemployment	Not Married
No access to a car	0.50	R **	0.68	0.52	0.52	0.59	0.54	0.58	0.58	0.53	0.51
Non-owner	0.84	0.90	R	0.87	0.84	0.91	0.87	0.88	0.85	0.91	0.91
Overcrowding	0.44	0.47	0.62	R	0.42	0.53	0.54	0.37	0.38	0.47	0.46
Low level of education	0.64	0.65	0.61	0.64	R	1.04	0.63	0.64	0.65	0.63	0.60
Unskilled worker	0.97	0.98	1.04	0.97	1.27	R	0.98	0.96	0.96	1.01	0.99
Foreign nationality	0.73	0.78	0.83	0.80	0.64	0.83	R	0.74	0.73	0.77	0.64
Single-parent household	1.11	1.16	1.27	1.08	1.11	1.10	1.13	R	1.01	1.13	1.28
Household with two or more persons	0.25	0.38	0.26	0.20	0.27	0.17	0.25	/ ***	R	0.18	0.45
Unemployment	0.97	0.99	1.12	0.98	0.95	1.04	0.99	0.97	0.95	R	1.03
Not married	0.39	0.40	0.57	0.40	0.33	0.43	0.34	0.54	0.49	0.46	R

* For the 10 alternative versions of F-EDI, only the variable removed is indicated; ** R: variable removed; *** With this alternative version of F-EDI, the variable was excluded from the model and, therefore, from the final score formula.

## Data Availability

The data presented in this study are available on request from the corresponding author.

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
