# Peer review of "Can an Ecological Index of Deprivation Be Used at the Country Level? The Case of the French Version of the European Deprivation Index (F-EDI)"

_ijerph, 2022, doi:10.3390/ijerph19042311_

Round 1

Reviewer 1 Report

ijerph-1575891

Title: How an aggregated index of deprivation fits the context: the case of the French version of the European Deprivation Index (F-EDI).

Type: Article

Dear Authors,

The submitted paper examines whether a national index can account for deprivation regardless of location characteristics, and calculates score deprivation for each IRIS using the French version of the European Deprivation Index (F-EDI). The 43,853 residential census block groups (IRIS) were divided into 8 types of living territories in Metropolitan France are used as the geographical scale.

The French version of the European Deprivation Index (F-EDI) is used in the title, but the ecological indices of deprivation appear many times in the manuscript. What is the relationship between F-EDI and ecological indices of deprivation?

The research background and literature review of the deprivation index need to be illustrated in more detail, in order to better justify the importance of the topic.

This manuscript needs to be supplemented with detailed data sources. Which institution published the original data, or what are other specific sources or websites? Corresponding references also need to be added.

Page 2 Line 91 Is the F-EDI updated in 2015 the latest version, as this may involve timeliness issues.

Page 2-3 Line 93-102 Whether the formula of F-EDI score is designed by the authors or refers to the existing research (if citing the work of others, please indicate the reference)? At least the formula of F-EDI does not reflect the innovation of this research. Then, what is the innovation of this study?

The selection of indicators and the determination of weights in the formula seem incomplete. If the scientificalness of the formula cannot be reflected, the empirical analysis in this manuscript may not be reliable.

Page 5 Line 165 The F-EDI score distribution in Metropolitan France (Figure 2) needs to be optimized. What method (Natural Breaks or others) is used for the classification of EDI score in the legend? It may better reflect the spatial characteristics by setting the number of classes to 7 with multi-color bands.

In addition, a more in-depth analysis of the spatial characteristics (spatial associations and spatial anomalies) of F-EDI can also be performed using the LISA method of ESDA.

In what concerns conclusions, an improvement is required. The theoretical and practical contributions should be described with more accuracy, as well as the study limitations.

Reviewer 2 Report

This is a really interesting paper and I thoroughly enjoyed reading it. It contains moderately surprising results with respect to the sub-domains of the F-EDI in different types of area of Metropolitan France, but it's main contribution will be to provide assurance of the general validity/utility of ecological deprivation indices in developed settings whilst also encouraging researchers to think more deeply about how and what they measure, and how this might vary by context. This is a very noble cause and has significant traction with public health research. Well done to these authors for conceiving and conducting the study, it will make a very useful addition to the literature and I look forward to citing this study :-)

I have some major comments which I think would improve the paper:

1) I think it would be worth the authors considering again the vocabulary around 'areas', 'territories', 'environments' etc. The terminology is not always consistent, and anyway I'm not so sure it's accurate in English. This may be an issue of translation - "Living Environments" sound a little like a biological eco-system, "Territory" normally has a political or administrative association. I think the different types of areas could be listed as just that - types of areas - or you could design/allocate a specific name to them and reference this early on and then use throughout e.g. "Area Classification" or similar. This even extends to the title which uses the idea of 'context'. I'm not sure that this is the right word here as it implies (at least in English) an implicit difference between the context of the thing and the thing itself, whereas here the context and the deprivation score itself are intrinsically linked to one another. See also the keywords for similar concerns.

2) Introduction line 45... 'being surrounded' rather than 'living surrounded by'... it also implies the person is not one of the disadvantaged people? This perhaps relates to point (1) - worth re-thinking exactly what is being said here?

3) Introduction para on EDI; I think this could be re-written to focus more on how this fits in with the current paper - there is an ideological jump between the previous and next paras as to the relevance of this content to the paper in hand.

4) I found the methodological description in lines 212-217 quite confusing. What has been done here exactly, and what then does Table 3 show? I'm sure I should be able to understand this perhaps from the Appendix Figures but it wasn't clear to me what this was doing.

5) This is probably beyond the scope of this paper, but it would be interesting to see the impact of the removal of each element on the differentials observed in an established outcome; for example, looking at mortality differences using the overall F-EDI and then comparing it to the 10 alternatives. Do some places suddenly get allocated to a much less deprived place? In which case what happens to the apparent 'inequality' in the outcome? There are clearly some quite notable outliers in terms of rank difference.

6) Did you consider removing 2 components? Again, potentially beyond the scope of the paper but something that could be considered or discussed.

7) Another idea (!) ... did you think, do you think there is room for a 'territory'(your word) adjusted F-EDI? Overall your data support it's general use as it is now but for more refined analyses, or policy-specific analyses should models take into account the territory classification? Or not? 

8) I'm quite surprised the paper does not consider a recent analysis by Fayet et al which uses novel methodology to generate a new classification of areas in France - interesting analyses and worth a read. How does this GeoCLASH compare to what the authors have used here? 

https://ij-healthgeographics.biomedcentral.com/track/pdf/10.1186/s12942-020-00242-0.pdf

Minor comments

a) I personally would have found the results section A LOT easier to read if you had replaced the labels of "Type 1, Type 2" etc with something more informative which reflects what these areas are actually like. This applies to all figures and tables throughout the submission.

b) The colours on the map Figure 2 are very similar - could you use a spectrum instead?

c) Introduction, first para, reference at the end of the last sentence would be good?

d) Review final lines of 'geographical scale' (lines 87-89) - is this really what you mean?

e) Provide the mean, sd of the areas you are using (lines 82-84). 

f) Line 157-9, do you mean contextual effects, plural?

g) Table 4 - add 'variable removed' on the diagonal (rather than white space)

h) Table 4 - mention that there are 10 alternative versions in the title of this table (potentially number the alternative models in the table too).

i) Table 4 - there is a stray '/' in the row household with 2 or more persons/single parent household col.

j) The labels a-i on Figures A1 and A2 are not helpful and could be replaced by descriptors.

k) There is no key provided for Figure A2. 

Reviewer 3 Report

An interesting paper on assessing deprivation indices to define social inequalities for health in France.

As it is important to provide the basis for comparisons, I would suggest to provide some additional information on the evidence provided through the use of EDI in other European countries both in the Introduction but most importantly in the Discussion section.

Also in the argumentation regarding the representativeness of the indices included e.g. owner of a car especially in combination with the rural/urban element, the authors should mention that an element of consideration is the presence of sustainable development practices or alternative means of eco-friendly transport.

Round 2

Reviewer 1 Report

Questions have been responded and revised. There are no further comments at this time.